# Mass Spectrometry-Based Workflow for the Identification and Quantification of Alternative and Canonical Proteins in Pancreatic Cancer Cells

**DOI:** 10.3390/cells13231966

**Published:** 2024-11-28

**Authors:** Clémence Guillon, Carole Pichereaux, Ikrame Lazar, Karima Chaoui, Emmanuelle Mouton-Barbosa, Mehdi Liauzun, Edith Gourbeyre, Pinar Altiner, David Bouyssié, Alexandre Stella, Odile Burlet-Schiltz, Serge Plaza, Yvan Martineau, Bertrand Fabre

**Affiliations:** 1Laboratoire de Recherche en Sciences Végétales (LRSV), CNRS/UT3/INPT, 31320 Auzeville-Tolosane, France; clemence.guillon@univ-tlse3.fr (C.G.); serge.plaza@univ-tlse3.fr (S.P.); 2Institut de Pharmacologie et de Biologie Structurale (IPBS), CNRS, UPS, Université de Toulouse, 31077 Toulouse, France; carole.pichereaux@ipbs.fr (C.P.); karima.chaoui@ipbs.fr (K.C.); emmanuelle.mouton-barbosa@ipbs.fr (E.M.-B.); pinar.altiner@ipbs.fr (P.A.); david.bouyssie@ipbs.fr (D.B.); alexandre.stella@ipbs.fr (A.S.); odile.schiltz@ipbs.fr (O.B.-S.); 3Fédération de Recherche (FR3450), Agrobiosciences, Interactions et Biodiversité (AIB), CNRS, 31326 Toulouse, France; 4Infrastructure Nationale de Protéomique, ProFI, FR 2048, 31077 Toulouse, France; 5MCD, Centre de Biologie Intégrative (CBI), CNRS, UT3, Université de Toulouse, 31400 Toulouse, France; ikrame.lazar@univ-tlse3.fr (I.L.); edith.gourbeyre@univ-tlse3.fr (E.G.); 6Centre de Recherche en Cancérologie de Toulouse (CRCT), INSERM U1037, Université Toulouse III-Paul Sabatier, ERL5294 CNRS, 31432 Toulouse, France; mehdi.liauzun@inserm.fr (M.L.); yvan.martineau@inserm.fr (Y.M.); 7Equipe Labellisée Ligue Contre Le Cancer, Université Toulouse III-Paul Sabatier, 31000 Toulouse, France

**Keywords:** alternative proteins, microproteins, proteomics, short open reading frame-encoded peptides, pancreatic ductal adenocarcinoma, data independent acquisition

## Abstract

The identification of small proteins and proteins produced from unannotated open reading frames (called alternative proteins or AltProts) has changed our vision of the proteome and has attracted more and more attention from the scientific community. Despite several studies investigating particular AltProts in diseases and demonstrating their importance in such context, we are still missing data on their expression and functions in many pathologies. Among these, pancreatic ductal adenocarcinoma (PDAC) is a particularly relevant case to study alternative proteins. Indeed, late detection of this disease, notably due to the lack of reliable biomarkers of early-stage PDAC, and the fact that tumors rapidly develop resistance to most of the treatments used in the clinics warrant the exploration of new repertoires of molecules. In the present article, we aim to investigate the alternative proteome of pancreatic cancer cell lines as a first attempt to decipher the expression of AltProts in PDAC. Thanks to a combined data-dependent and data-independent acquisition mass spectrometry workflow, we were able to identify tryptic peptides matching 113 AltProts in a panel of 6 cell lines. In addition, we identified AltProts differentially expressed between pancreatic cancer cell lines and other cells (HeLa and HEK293T). Finally, mining the TCGA and Gtex databases showed that the corresponding transcripts encoding several AltProts we identified are differentially expressed between PDAC tumors and normal tissues and are correlated with the patient’s survival.

## 1. Introduction

Pancreatic ductal adenocarcinoma (PDAC) is the fourth leading cause of cancer-related mortality and is expected to become the second-leading cause of cancer death in the United States of America by 2030 [1]. Despite its low incidence, 12 per 100,000 per year, and a 1.3% risk of developing the disease in a lifetime, the 5-year survival rates remain as low as 3–15% [2]. Tumors rapidly develop resistance to most of the treatments used in the clinics [3]. Surgery is still the best available therapeutic option and is particularly successful for the early stages of pancreatic cancer. However, most PDAC cases are diagnosed at advanced stages so only 10 to 20% of tumors are eligible for surgery. Indeed, in the United Kingdom, 85% of patients diagnosed with this cancer, due to the apparition of symptoms associated with PDAC, cannot be treated by surgery anymore [4]. Despite an important effort from the scientific community to identify reliable biomarkers and new drugs (hundreds of published articles on the topic), only a few candidates have translated to clinical use for the early detection of PDAC [5]. Most of the studies focused on the identification of differential abundance of metabolites or proteins annotated in common databases (e.g., UniprotKB for proteins) in the serum of PDAC patients. It is thus necessary to consider alternative repertoires of molecules other than monitoring and targeting annotated proteins.

The discovery of the pervasive expression of small proteins and proteins produced from unannotated open reading frames (ORFs) has changed our vision of the proteome. Indeed, it is now well established that thousands of ORFs across the kingdom of life were missed during genome annotations and actually coded for stable proteins [6]. These proteins are called alternative proteins (AltProts), short open reading frame-encoded peptides (SEPs), or microproteins. Alternative proteins can be produced from predicted long non-coding RNA (lncRNA), the 5′ and 3′ untranslated region (UTR) of mRNA, or even from coding sequences (CDS) but from another reading frame [7]. Although databases such as OpenProt [8], sORF.org [9], and others [10,11] provide tens of thousands of possible unannotated ORFs encoding for alternative proteins in different species, it is still not clear which ones are genuine. The development of approaches such as ribosome profiling and mass spectrometry has greatly helped to detect new ORFs and AltProts, respectively, in recent years [12]. Despite important advances in the field, identifying AltProts still remains challenging due to the need for dedicated biochemical enrichment methods, curated databases, and specific bioinformatics analysis [12]. Notably, there is still no unified approach or software dedicated to the identification of AltProts, which are shorter than canonical proteins on average, thus, generating fewer peptides following enzymatic digestion (often only one peptide) [12]. To prove the production of an AltProt, it is necessary to use more stringent criteria than for typical proteomic approaches as well as ensuring that the peptides measured by MS are proteotypic [12]. Although the roles of most AltProts remain elusive, more and more research groups focus their attention on deciphering their functions [13,14]. These small proteins seem to have a large spectrum of roles in many key cellular processes [13,14]. Another question remaining is the implication of AltProts in pathologies. Several studies have investigated particular AltProts in diseases and demonstrate their importance in such context [13,15]. Notably, it is now established that AltProts are involved in most hallmarks of cancer [15]. However, data on their expression in the different cancer types are still lacking.

In the present article, we investigated the alternative proteome of pancreatic cancer cell lines as a first attempt to decipher the expression of AltProts in PDAC. We developed a combined data-dependent (DDA) and data-independent (DIA) acquisitions mass spectrometry workflow in order to identify and quantify AltProts, as well as canonical proteins. Thanks to that approach, we were able to identify tryptic peptides matching 113 AltProts in a panel of 6 cell lines. We showed that 60% of these AltProts harbor at least one known protein domain and are predicted to have predominantly a nucleocytoplasmic localization. In addition, we identified AltProts that seem to be differentially expressed between pancreatic cancer cell lines and other cells (HeLa and HEK293T cell lines). Finally, mining the TCGA and Gtex databases showed that the corresponding transcripts (annotated as “non-coding RNAs”) encoding several AltProts we identified are differentially expressed between PDAC tumors and normal tissues. Interestingly, the expression of these “non-coding RNAs” is correlated with the patient’s survival. Finally, we were also able to validate previously published data suggesting that several canonical proteins could be used as biomarkers in PDAC.

## 2. Materials and Methods

### 2.1. Cell Culture

PANC-1, MIA PaCa-2, AsPC-1, and PATU8988T pancreatic cancer cell lines and HeLa and HEK293T cells were grown as described previously [16,17]. Four biological replicates were used for subsequent experiments for all the cell lines except PATU8988T (3 biological replicates).

### 2.2. Sample Preparation

Cell lysis was performed using a lysis buffer containing ammonium bicarbonate 50 mM and sodium deoxycholate 1%. The samples were sonicated, boiled for 5 min at 95 °C, and centrifugated for 10 min at 14,000× *g*. The pellets were discarded. Protein concentration was measured using a BCA protein assay (Pierce™, Waltham, Massachusetts, USA). About 200 µg of proteins were reduced with 10 mM DTT for 30 min at 37 °C and alkylated with 30 mM chloroacetamide for 30 min at RT in the dark. Then, the samples were diluted 5 times with 50 mM ammonium bicarbonate and trypsin was added (trypsin/protein ratio of 1/50) for O/N digestion. The next day, the samples were acidified with trifluoroacetic acid to a final concentration of 0.4% and centrifuged at 14,000× *g* for 10 min. The pellets were discarded. The peptides were desalted on a C18 column (Pierce™) and dried down using a speedvac. Half of the tryptic peptides were kept for data-independent analysis and the other half was used for high pH reversed-phase fractionation (Pierce™ High pH Reversed-Phase Peptide Fractionation Kit) according to the manufacturer’s instructions. For each cell line, all the biological replicates were combined (same quantity of peptides from each replicate) to obtain 100 µg of peptides, which were fractionated. Each fraction (5 per cell line) was dried down using a speedvac.

### 2.3. Data-Dependent Acquisition (DDA) Mass Spectrometry Analysis

Tryptic peptides from each fraction from the high pH reversed-phase separation were resuspended in 0.2% formic acid at a concentration of 500 ng/µL and analyzed by nano-LC-MS/MS using a nanoRS UHPLC system coupled to an Orbitrap Exploris 480 mass spectrometer using FAIMS Pro Duo interface (Thermo Fisher Scientific, Bremen, Germany). One microliter of each sample was loaded on an analytical C18 column (PEPMAP C18 2 µm 75 µm × 500 mm Thermo Fisher Electron) heated at 45 °C. The mobile phase flow rate was set to 300 nL/min, and 5% ACN + 0.2% formic acid (FA) in H_2_Omq and 80% ACN + 0.2% FA in H_2_Omq were used as buffers A and B, respectively. Peptides were eluted using a 95 min gradient: 0–25% buffer B for 58 min, 25–40% buffer B for 20 min, 40–90% buffer B for 2 min, and 90% buffer B for 5 min. The Orbitrap Exploris 480 was operated in FAIMS mode (gas flow of 3.9 L/min, two compensation voltages used: −45 and −60 v, 0.8 and 0.7 s cycle times per CV) with the Xcalibur software. Survey scan MS spectra were acquired in the Orbitrap on the 375–1200 *m*/*z* range with the resolution set to a value of 60,000; the AGC target was 300% with a maximum injection time of 50 ms. Following each survey scan, the most intense ions above a threshold ion count of 5 × 10^3^ were selected for fragmentation by high-energy collision-induced dissociation at a normalized collision energy of 30%. The number of selected precursor ions for fragmentation was determined by the “Top Speed” acquisition algorithm. AGC target value was set at 100% and automatic maximum injection time. The MS2 resolution was set to 15,000. Isolation width was set at 1.6 *m*/*z*, and dynamic exclusion was used within 45 s to prevent repetitive selection of the same peptide.

Raw data were analyzed using a combination of Mascot (version 2.8), Percolator, and Proline (version 2.1) [18]. Mascot was used as the search engine with default parameters, N-terminal acetylation and methionine oxidation as variable modifications, and carbamidomethylation of cysteine as fixed modification. A database is constituted of the AltProts and Isoforms of OpenProt (version 1.6), proteins from Uniprot (June 2023, 81,791 sequences), and common contaminants. Obtained PSMs were rescored using Percolator before import in the Proline datastore. Imported search results were subsequently validated into Proline with a final FDR of 1% and 5% at the PSM and protein levels, respectively. Then, all the PSMs matching AltProts or Isoforms were filtered to keep only the PSMs for which the y or b ions covered at least 75% of the peptide sequence. The resulting PSMs were manually validated by two operators. Validated peptides were checked for proteotypicity using the Peptide search module from Uniprot (against all proteomes). A spectral library containing the annotated MS/MS spectra of the resulting AltProts and canonical proteins was generated using Proline. An R script has been applied to make the spectral library compliant with the DIA-NN format (script provided as Appendix A).

### 2.4. Data-Independent Acquisition (DIA) Mass Spectrometry Analysis

Non-fractionated tryptic peptides were resuspended in 0.2% formic acid at a concentration of 500 ng/µL and analyzed on the same MS platform as for the DDA analysis. One microliter of each sample was loaded on an analytical C18 column (PEPMAP C18 2 µm 75 µm × 500 mm Thermo Fisher Electron) heated at 45 °C; 1 µL of each sample was injected on the analytical C18 column (75 μm inner diameter × 50 cm, Pepmap C18, 2 µm, Thermofisher Electron SAS) equilibrated in 95% solvent A (5% acetonitrile, 0.2% formic acid) and 5% solvent B (80% acetonitrile, 0.2% formic acid). Peptides were eluted using a 95 min gradient: 0–25% buffer B for 58 min, 25–40% buffer B for 20 min, 40–90% buffer B for 2 min, and 90% buffer B for 5 min. The mass spectrometer was operated in data-independent acquisition mode with the Xcalibur software. Full-scan MS resolution was set at 60,000, and the AGC target was 300% with a maximum injection time of 100 ms. The MS1 mass range was set to 400–1008. For MS2 spectra, AGC target value spectra were set at 1000%. The DIA isolation scheme consisted of 2 × 75 staggered windows (8 *m*/*z* width, 4 *m*/*z* offset) covering the 400–1000 *m*/*z* range (the exact *m*/*z* values are provided in Appendix A). The resolution was set to 15,000, injection time was set in automatic mode, and normalized collision energy was set at 30%.

Raw data were processed using DIA-NN (version 1.8) [19] using default parameters and the spectral library generated with the DDA analysis. DIA-NN reports were post-processed using the Proline software v2.2: norm. No imputation of missing values was performed. Quantitative data were inspected for each AltProt. All the *p*-values were obtained using Welch’s *t*-test.

### 2.5. Data Availability

All the mass spectrometry data have been deposited with the MassIVE repository with the dataset identifier: MSV000095914.

### 2.6. Alternative Proteins Analysis

InterPro [20] was used for protein domain and disordered region identification on alternative proteins.

SLiMs identification within AltProts and isoforms was performed using the ELM prediction tool [21] from the Eukaryotic Linear Motif resource without specified cellular compartment, *Homo sapiens* as the Taxonomic Context and Motif Probability Cutoff of 100.

Protein localization was predicted using DeepLoc 2.0 [22].

The prediction of transmembrane helices was performed using TMHMM 2.0 [23].

The data and transcript accessions about the RNAs encoding AltProts were retrieved from OpenProt (https://archive.openprot.org/).

Mining the TCGA and Gtex databases was performed using GEPIA2 [24] as described previously [25], except that quartile for the Group Cutoff and PAAD for the Datasets Selection were used for survival analysis (all other parameters were set by default).

## 3. Results and Discussion

### 3.1. Development of a Mass Spectrometry-Based Workflow for the Identification and Quantification of Alternative Proteins in Pancreatic Cancer Cell Lines

In order to explore the alternative proteome of pancreatic cancer cells, we developed a workflow using pancreatic cancer cell lines in culture as starting material (Figure 1). We used HeLa (Cervix adenocarcinoma) and HEK293T (human embryonic kidney immortalized cells) cell lines in order to decipher AltProts expressed in pancreatic cancer cell lines. High pH reverse phase fractionation was used following cell lysis and protein digestion to improve AltProts identification (Figure 1). In addition to the identification of alternative proteins, this workflow also enables the identification of canonical proteins since we did not perform any particular depletion of large proteins (Figure 1). Fractionated samples were analyzed using data-dependent acquisition (DDA) on a Thermofisher Exploris 480 instrument (Figure 1). Unfractionated samples were analyzed in parallel in data-independent acquisition (DIA) mode on the same instrument (Figure 1). Using the OpenProt database [7] and Proline [18] for the analysis of the DDA data, as well as manual validation of the peptide spectrum matches (PSMs), 113 AltProts were identified (Figure 1 and Appendix A). Particular attention was put to the validation of AltProts’ PSMs (Figure 1). To maximize our chances of avoiding false positive identification, the y or b ions should cover at least 75% of the peptide sequence, and each PSM for AltProts was manually validated by two operators. Then, all the validated tryptic peptides matching for AltProts were blasted against all the proteomes in Uniprot to ensure their proteotypicity. In addition, nearly 10,000 canonical proteins were also identified across all the cell lines analyzed (Figure 1 and Appendix A), highlighting the good analytical depth of our approach. A spectral library based on these data was generated in order to analyze the DIA data (Figure 1). The MS-based approach developed here thus enables the identification and quantification of alternative proteins, as well as canonical proteins.

### 3.2. Structural Properties of the Alternative Proteins Identified

Next, we looked at the different properties of the alternative proteins identified. Nearly 80% of the alternative proteins identified are 150 amino acids long or less and their measured median size was 84 amino acids (Figure 2A and Appendix A). The isoelectric point (pI) of the alternative proteins identified spanned from 3.9 to 13 with a median of 9.07 and 60% of them had a pI between 8 and 12 (Figure 2B and Appendix A). Looking at the different types of RNA from which the alternative proteins identified in this study are produced, it seems that most of them (72%) are translated from RNAs that were predicted to be non-coding (lncRNAs) (Figure 2C) and more particularly from pseudogenes (Appendix A). Only 18% of alternative proteins are produced from new ORFs on mRNA (Figure 2C and Appendix A). The proportion of AltProts produced from lncRNAs in this study is higher than previous reports in other human cell lines (54%) [26]. If we compare the data from the ones obtained in a previous study in Drosophila melanogaster, we can see a clear difference in the source of production of AltProts between the two species as nearly 80% of AltProts identified in flies were expressed from new ORFs on mRNA, whereas only 13% were produced from lncRNA [27]. This was also different from a study in mice in which the lncRNAs (52%) and mRNAs (48%) were nearly equally distributed as AltProts production sources [28]. These data point towards different main sources of production of unannotated proteins in different species.

Next, we looked for the presence of known protein domains based on the sequences of identified AltProts. First, an analysis using InterPro revealed that 41% of the identified AltProts bear at least one disordered region (Figure 3A and Appendix A). We could also observe that more than 60% of the identified AltProts have at least one known protein domain (Figure 3B and Appendix A). Diving more into the data showed that the protein domains identified are linked to the ribosome, RNA-binding, GAPDH, ubiquitin–proteasome pathway, or immunoglobulins and major histocompatibility complex (Figure 3C and Appendix A). The presence of short linear motifs (SLiMs) was also investigated. SLiMs are involved in protein–protein interactions and located in disordered regions of proteins [30]. Importantly, the majority of the SLiMs identified are linked to intracellular signaling pathways (54.3% of the SLiMs identified), and many SLiMs associated with the ubiquitin–proteasome or SUMO pathways (12.4% of the SLiMs identified) were also present on AltProts (Figure 3D and Appendix A). Finally, we used DeepLoc to predict the subcellular localization of AltProts. For most of the AltProts, a cytosolic (54%) or nuclear (29%) localization was predicted (Appendix A). However, mitochondrial (9%) and extracellular (6%) or membrane (2%) localizations were also predicted (Appendix A). The presence of transmembrane helices was confirmed for the AltProts predicted to be membrane proteins using TMHMM 2.0 (Appendix A). All in all, these analyses point towards possible functions of the alternative proteins identified in our study.

### 3.3. Quantitative Analysis of the Alternative Proteome in Pancreatic Cancer Cell Lines

Thanks to our experimental workflow, it was possible to obtain quantitative data from DIA for 55 of the alternative proteins identified (Figure 4A and Appendix A). The intensity measured for each of these AltProts was compared between pancreatic cancer cell lines and HeLa and HEK293T (Figure 4A). In total, we found 11 AltProts that are differentially expressed between pancreatic cancer cell lines and the two other cell types (Figure 4B–D, Appendix A). For six of these cases, the AltProt was either found only in pancreatic or only in other cell lines (Figure 4D, Appendix A). Eight AltProts were found more abundant in pancreatic cancer cell lines (Figure 4C,D, Appendix A). The expression of each AltProt was also monitored for each pancreatic cancer cell line as compared to HeLa and HEK293T. We could observe that certain AltProts are expressed mainly in one particular cell line (Appendix A). The higher abundance of these AltProts in pancreatic cancer cell lines could have different explanations. First, this high abundance could be due to the high expression of these AltProts in pancreatic tissues or derived cells. Alternatively, these AltProts might be more expressed specifically in pancreatic cancer cells/tumors. In any case, further experiments will be necessary to define if the identified AltProts as more expressed in pancreatic cancer cells could be used for the diagnosis of the pathology and/or if they have a role in tumorigenesis in this type of cancer.

### 3.4. RNA-Encoding Alternative Proteins Are Differentially Expressed in Pancreatic Ductal Adenocarcinoma and Their Levels Correlate with Patients’ Survival

The online tool GEPIA2 was used to mine the TCGA and Gtex data about RNA expression and their correlation with patient survival in PDAC [24]. We could retrieve RNA expression data for 95 out of the 113 identified in this study and 44 of them could be used for survival analysis. Looking at the expression of RNA-encoding AltProts, five of them displayed a significant correlation between their expression level and the survival of PDAC patients (Figure 5A–C and Appendix A). In three of these cases, a higher expression of the RNA encoding the AltProts IP_653613, IP_747506, and IP_150823 correlated with a decreased patient survival (hazard ratio of 2, 2.2, and 2.8, respectively) (Figure 5A–C). For the AltProts IP_691622 and IP_752246, higher expression of their corresponding RNAs is a good prognostic indicator for the patients (Appendix A). Of note, three of the five AltProts displaying a significant correlation between expression level and patient survival are encoded by predicted long non-coding RNAs (IP_691622, IP_747506, and IP_653613), while the other two are encoded on mRNAs (IP_150823 and IP_752246). In the case of AltProts encoded by mRNAs, it is difficult to infer the difference in patient survival to the AltProt or the canonical protein. However, in the case of AltProts encoded by lncRNAs, it might point toward a role of these AltProts in PDAC biology and warrant further investigation of these AltProts in this cancer type. In addition, we also looked for a correlation between the RNAs encoding the AltProts identified in this study in other cancer types.

Interestingly, we found correlations between the expression levels of AltProt’s RNAs and patient survival in different cancer types for many of the AltProts identified (Appendix A). As an example, it was very noticeable for Kidney Renal Clear Cell Carcinoma (KIRC) for which high or low expression levels of RNAs coding for 18 of the AltProts impact patient survival (Appendix A).

### 3.5. Quantitative Analysis of the Canonical Proteome in Pancreatic Cancer Cell Lines

In addition to the identification and quantification of AltProts, our MS-based workflow enables to monitoring of changes in the expression of canonical proteins. We notably compared the abundance for each protein quantified between the panel of pancreatic cell lines and HeLa and HEK293T cells (Figure 6A and Appendix A). Out of the 9993 proteins quantified using DIA, 658 proteins were more abundant in HeLa and HEK293T cells, whereas 821 proteins were more abundant in pancreatic cancer cell lines (Figure 6A and Appendix A). Pathways such as cell cycle, DNA replication, or autophagy were enriched in HeLa and HEK293T cells versus pancreatic cell lines (Figure 6B).

On the other hand, proteins belonging to ribosome biogenesis, cell–cell junction organization, or fatty acid oxidation were more abundant in pancreatic cancer cell lines (Figure 6C). As an example, the proteins BOP1, DDX21, DDX27, EMG1, ISG20, MPHOSPH10, NAT10, NGDN, NOL6, NOP58, NSUN5, RPF1, RPF2, RRP12, RRP9, RRS1, UTP14A, UTP23, UTP4, WDR12, and WDR43, which are rRNA modifications enzymes and structural components of pre-ribosomal sub-complexes [31], were found more abundant in pancreatic cancer cell lines. It is well known that mRNA translation is dysregulated in pancreatic cancer cells [32]. In addition, a growing body of data links ribosome biogenesis as a potential target for pancreatic cancer treatment [33,34]. Of note, as for the expression of AltProts, these data reveal that these proteins/pathways might be more abundant in pancreatic cancer cells or in normal pancreas tissue. Interestingly, several proteins were found more abundant in all the pancreatic cancer cell lines versus HeLa and HEK293T (Appendix A). As an example, the proteins NT5E and RHPN2 were found highly expressed in pancreatic cancer cell lines (Figure 7A,B). Thanks to an analysis with GEPIA2, the mRNAs coding for these proteins were found to be more expressed in PDAC tumors versus normal tissues (Figure 7C,D), and high expression of these genes correlates with poor prognosis of PDAC patients (Figure 7E,F). NT5E was recently identified as a novel prognostic biomarker in cancer-associated fibroblasts notably in PDAC [35]. Regarding RHPN2, another recent study has shown that overexpression of this protein promotes PDAC [36]. Our data corroborate the potential role of these proteins in PDAC biology. In addition to NT5E and RHPN2, we also found the proteins APOE, APOL1, and TIMP1, which were previously described as possible biomarkers for PDAC [37,38]. However, each of these proteins was found to be increased only in one cell line, the PANC1, PATU8988T, or MIAPAC2 cell line, respectively (Appendix A). Another surprising result was the proteins S100A4 and S100A6 that were expressed in all the cancer cell lines analyzed here but not in HEK293T cells (Appendix A). Both proteins are more expressed in PDAC tumors, and high expression in PDAC patients correlates with poor prognosis (Appendix A). These proteins were also proposed as potential biomarkers in pancreatic cancer [39].

## 4. Conclusions

In the present article, we present an initial investigation of the alternative proteome of pancreatic cancer cells. We developed an integrated mass spectrometry-based workflow combining data-dependent and -independent acquisitions to identify and quantify alternative and canonical proteins in one analysis (Figure 1). Thanks to this approach, we were able to identify a set of alternative proteins that might have an important role in pancreatic ductal adenocarcinoma biology. Indeed, the expression of RNAs coding for some of the alternative proteins identified here is altered in PDAC tumors versus normal tissues and correlates with patient survival (Figure 5 and Appendix A). We were also able to corroborate previously published data pointing towards the use of several canonical proteins as biomarkers in PDAC (Figure 7 and Appendix A).

The present work provides new data about the expression of alternative proteins in PDAC and serves as a foundation for future studies investigating the alternative proteome in pancreatic tumors or other cancer types.

## Figures and Tables

**Figure 1 cells-13-01966-f001:**
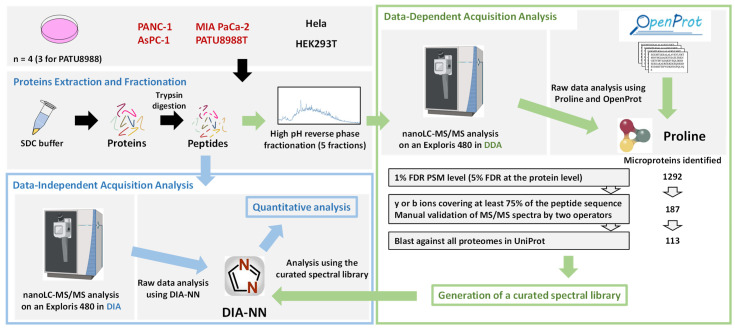
Mass spectrometry-based workflow to identify alternative proteins in pancreatic cancer cell lines. Proteins were extracted from pancreatic or HeLa and HEK293T cell lines, digested with trypsin and most of the resulting peptides were fractionated using a high pH reverse phase. The resulting fractionated peptides were analyzed in data-dependent acquisition (DDA) mode, and the generated raw data were processed using Proline with the OpenProt database to identify alternative and canonical proteins. Peptide spectrum matches were manually validated, and a curated spectral library was created. In parallel, unfractionated peptides were analyzed in data-independent acquisition (DIA) mode, and the generated raw data were processed using DIA-NN and the curated spectral library created using DDA data.

**Figure 2 cells-13-01966-f002:**
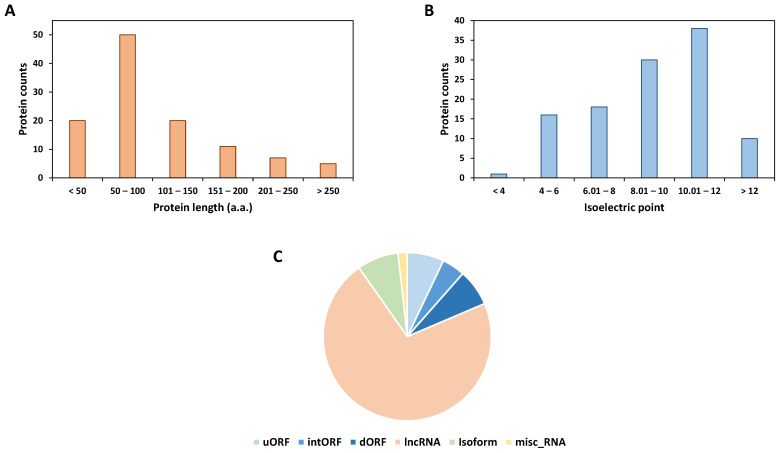
Distribution of amino acid length, isoelectric point, and source of production of the alternative proteins identified. (**A**) Distribution of the amino acid length of the AltProts identified in this study. (**B**) Repartition of AltProts depending on their isoelectric point. (**C**) Proportion of alternative proteins produced from ORFs from novel isoforms, ORFs located in the 5′UTR (uORF), coding sequence (intORF), or 3′UTR (dORF) of mRNAs, from non-annotated ORFs on long non–coding RNAs (lncRNA) or miscellaneous RNA (misc_RNA). The nomenclature of the different classes of ORFs was adapted from [29].

**Figure 3 cells-13-01966-f003:**
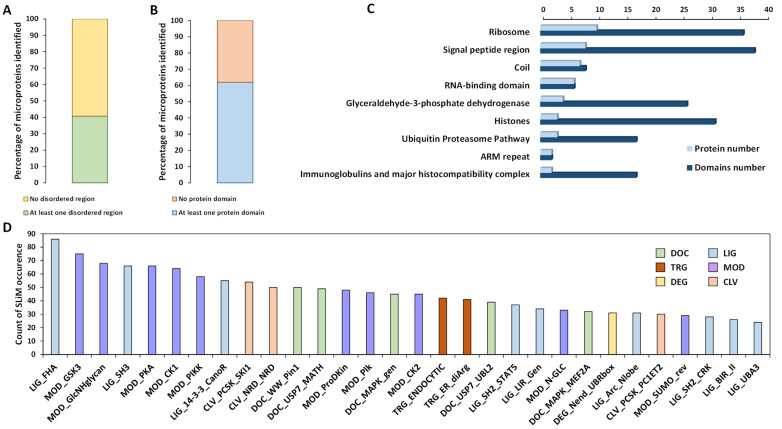
Identification of protein domains and short linear motifs in alternative proteins. (**A**) Proportion of AltProts with at least one predicted disordered region predicted by InterPro. (**B**) Proportion of AltProts with at least one known protein domain identified by InterPro. (**C**) Frequency of protein domains and number of proteins bearing a particular protein domain in the AltProts identified. (**D**) Counts of the different classes of SLiMs identified from the AltProts sequence. SLiM classes are targeting sites for subcellular localization (TRG), post-translational modification sites (MOD), ligand-binding sites (LIG), docking sites (DOC), degradation sites (DEG), and proteolytic cleavage sites (CLV).

**Figure 4 cells-13-01966-f004:**
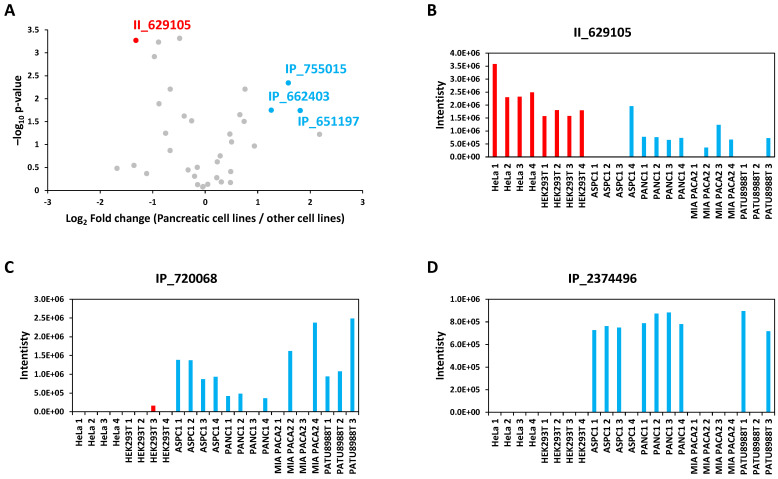
Quantitative analysis of the data-independent acquisition of the alternative proteins between pancreatic cancer cells and HeLa and HEK293T. (**A**) Volcano plot representing the log 2 ratio (pancreatic cell lines/other cell lines) for each AltProt quantified in the DIA analysis and the corresponding Welch’s *t*-test *p*-value (−Log_10_ transformed). The blue dots represent the AltProts more abundant in pancreatic cancer cell lines (*p*-value < 0.05 and Log_2_ fold change > 1), red dots represent the AltProts more abundant in HeLa and HEK293T cells (other cell lines) (*p*-value < 0.05 and Log_2_ fold change < 1), and gray dots represent AltProts not differentially expressed between pancreatic cancer cell lines and other cell lines. (**B**–**D**) Chart displaying the protein intensity measured for the AltProts II_629105 (**A**), IP_720068 (**B**), and IP_2374496 (**C**) across all the replicates of all the cell lines used in this study.

**Figure 5 cells-13-01966-f005:**
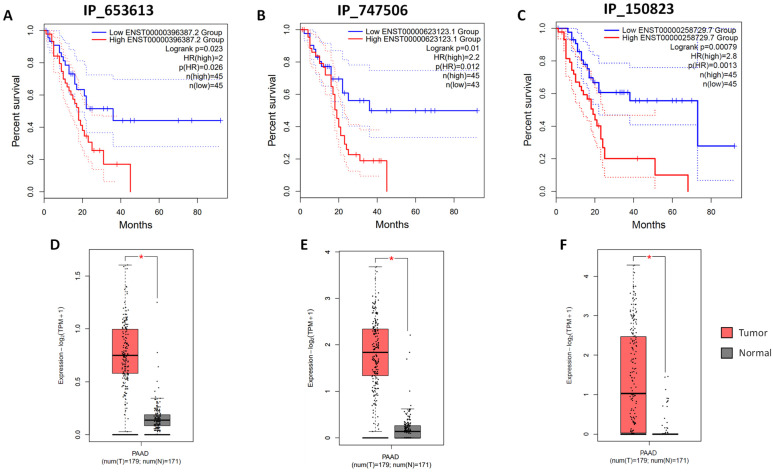
Several RNAs coding for alternative proteins are differentially expressed between tumors and normal tissues and their level correlates with PDAC patient’s survival. (**A**–**C**) Survival analysis of patients expressing high and low levels of RNAs encoding the IP_653613 (**A**), IP_747506 (**B**), and IP_150823 (**C**) AltProts. (**D**–**F**) Expression levels of the RNAs encoding the IP_653613 (**A**), IP_747506 (**B**), and IP_150823 (**C**) AltProts in PDAC (PAAD) and corresponding normal pancreatic tissues (based on TCGA and Gtex data). The red * is displayed for *p*-values < 0.01.

**Figure 6 cells-13-01966-f006:**
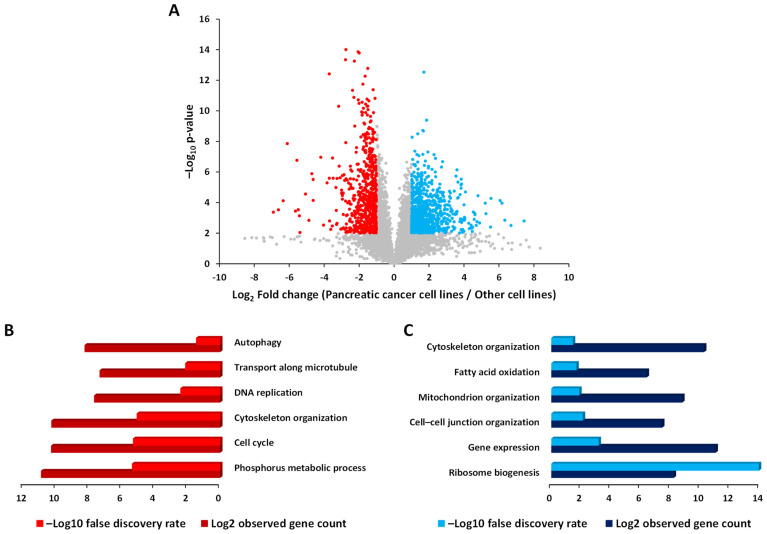
Comparison of the proteome from HeLa and HEK293T versus pancreatic cancer cell lines. (**A**) Volcano plot representing the log_2_ ratio (Pancreatic cancer cell lines/Other cell lines) for each protein quantified and the corresponding −Log_10_ *p*-value. The blue, red, and gray dots represent the proteins more abundant in the pancreatic cancer cell lines (*p*-value < 0.01 and Log_2_ fold change > 1), more abundant in the other cell lines (HeLa and HEK293T) (*p*-value < 0.01 and Log_2_ fold change < −1), or not differentially expressed, respectively. (**B**,**C**) Graphs displaying the pathways identified in a genome ontology (GO) analysis from the proteins more abundant in HeLa and HEK293T (**B**) or more abundant in pancreatic cancer cell lines (**C**). The number of genes (Log_2_ transformed gene count) and the false discovery rates (FDR, −log_10_ transformed) are displayed in the figures. Higher values indicate a higher number of genes identified from a pathway and a better FDR value.

**Figure 7 cells-13-01966-f007:**
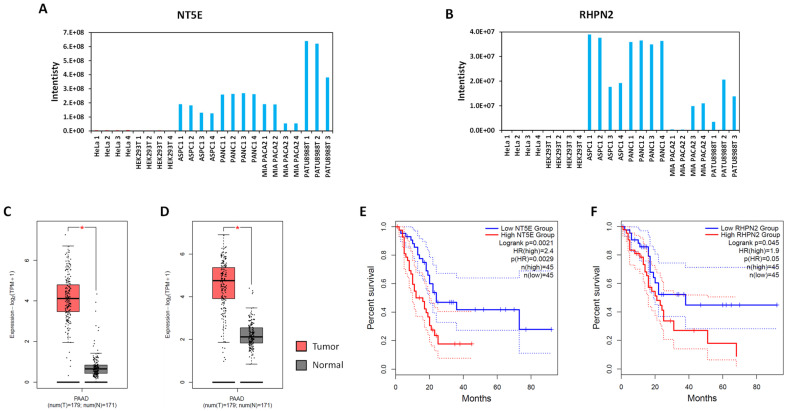
Several canonical proteins are more expressed in cancer pancreatic cell lines and their corresponding mRNAs are differentially expressed between tumors and normal tissues and their level correlates with PDAC patient’s survival. (**A**,**B**) Chart displaying the protein intensity measured for the NT5E (**A**) and RHPN2 (**B**) proteins. (**C**,**D**) Expression levels of the mRNAs encoding the NT5E (**C**) and RHPN2 (**D**) proteins in PDAC (PAAD) and corresponding normal pancreatic tissues (based on TCGA and Gtex data). The red * is displayed for *p*-values < 0.01. (**E**,**F**) Survival analysis of PDAC patients expressing high and low levels of mRNAs encoding the NT5E (**E**) and RHPN2 (**F**) proteins.

## Data Availability

All the mass spectrometry data have been deposited with the MassIVE repository with the dataset identifier: MSV000095914.

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
