# Peer review of "Mass Spectrometry-Based Workflow for the Identification and Quantification of Alternative and Canonical Proteins in Pancreatic Cancer Cells"

_cells, 2024, doi:10.3390/cells13231966_

Round 1

Reviewer 1 Report

Comments and Suggestions for Authors

Thank you for allowing me to review your manuscript. Please find my comments below.

Summary

In this study the authors tried to decipher the expression of AltProts in PDAC. To achieve this, they used a combined data dependent and independent mass spectrometry analysis of pancreatic cancer cell line proteome and compared it to other cell lines. They also mined the TCGA and Gtex datasets for the corresponding transcripts encoding several AltProts. They found 113 AltProts in a panel of 6 cell lines. They also found Altprots that were differentially expressed when compared to other cells lines. mRNA transcripts encoding several Altprots were identified on the TCGA and Gtex databases which were differentially expressed in PDAC tumors versus normal tissues, and they also correlated with patient survival.

Originality

This is an original study which discusses the role of AltProts in pancreatic cancer diagnosis and prognosis.

Strengths

This is an original study which shines light on an important field of pancreatic cancer diagnosis and prognoss. The methodology appears to be appropriate, and the manuscript is well written.

Weaknesses

I think comparing the pancreatic cancer cell lines with other cancer cell line (rather than the combined cancer and other cell line which was used in this study) would have given us information about altprots unique to this cancer. Also, comparison with a normal pancreas cell line would have shown proteins unique to pancreatic cancer.

Other comments are as below.

Methodology

1.       Why were both data dependent and independent analytical methods employed?

2.       Please describe how the RNA from which the altprots was translated identified?

3.       Has there been any other study which has compared the mRNA source of altprots in different species?

Results

1. Can you compare the abundance of altprots in pancreatic cancer cells lines versus HeLa and HEK293T separately without combining them as one is a cancer cell line and the other is not?

2. How many transcripts matching the AltProts discovered in the PDAC cell lines were seen in the TCGA an Gtex datasets?

Reviewer 2 Report

Comments and Suggestions for Authors

The manuscript by Guillon et al., “Mass spectrometry-based workflow for the identification and quantification of alternative and canonical proteins in pancreatic cancer cells” identified transcripts that encode AltProts which are differentially expressed between pancreatic cancer cell lines and other cells (HeLa and HEK293T). These AltProts fall into four classes of major interest; ORFs located in the 5′UTR (uORF), 3′UTR (dORF), lncRNA, and misc_RNA, which are presumably transcripts from unannotated regions of the genome. Some of these AltProts transcripts correlated with patient survival and are differentially expressed between PDAC tumors and normal tissues. This is an important but descriptive study. The interesting implications of the expression of neoantigens in PDA are not investigated. Furthermore, the association of these AltProts with a certain type of PDA type, such as classical or basal-like/squamous,  was not investigated.     

1.     Reviewer Comment: Figure 4A. Indicate which blue dots are IP_653613 (A), IP_747506 (B) and 320 IP_150823 (C) in the log2 fold change plot of Panc cell lines/other cell lines.  Given that RNAs for these are expressed in human PDA tumors (Fig. 5), are these AltProts proteins also expressed in human PDA tumors? What are the implications as targets for immunotherapy?  Are these RNAs and/or proteins co-expressed with other markers? 

2.     Reviewer Comment: Lines 341-345: This is a common finding (i.e. “ … correlations between the expression levels of RNAs and patient survival in different cancer types …”). Can the authors add anything more to this descriptive aside? 

3.     Reviewer Comment: Figure 3A: What is the significance of finding disordered regions in the short peptides – might they be useful targets for therapy?

4.     Reviewer Comment: Lines 361-369: Rewrite the paragraphs to put information about the 821 proteins in the paragraph about Ribsome biogenesis, etc. (bold underline below)

Out of the 9,993 proteins quantified using DIA, 658 proteins were more abundant in HeLa and HEK293T cells whereas 821 proteins were more abundant in pancreatic cancer cell lines (Figure 6A and Supplementary table 3). Pathways such as Cell cycle, DNA replication or 364 Autophagy were enriched in HeLa and HEK293T cells versus pancreatic cell lines (Figure6B). 

On the other hand, proteins belonging to Ribosome biogenesis, Cell-cell junction organization or Fatty acid oxidation were more abundant in pancreatic cancer cell lines (Figure 6C). 

5.     Reviewer Comment: How many of the 821 proteins that were more abundant in pancreatic cancer cell lines have been previously reported as markers for PDA?

Minor points:

6.     Reviewer Comment: I could not find legends for the Supp Figs.

7.     Reviewer Comment: Typo:  “… AltProts bear at least on_ disordered region …” 

add e, “one” 

8.     Reviewer Comment: Insert missing word (line 298):  First, this high abundance could be due to high expression of these AltProts in pancreatic tissues or derived cells. 

9.     Reviewer Comment: Insert missing word (line 300):  “Alternatively, the expression of these AltProts might be more expressed in (…missing words …).  In any case, further experiments will be necessary …”

10.  Reviewer Comment: Table A2 was not in the manuscript or Supp Tables provided for review. It is unlikely to be in ref 24.

Line 324:  “Table A2 was used to mine the TCGA and Gtex data about RNA expression and their correlation with patient survival in PDAC [24].” 

11.  Reviewer Comment: Line 394: suggestion - replace “first” with “initial”;   … we present an initial firstinvestigation …” 
